# Optical Rotation—A Reliable Parameter for Authentication of Honey?

**DOI:** 10.3390/molecules27248916

**Published:** 2022-12-15

**Authors:** Dessislava Gerginova, Vanya Kurteva, Svetlana Simova

**Affiliations:** 1Bulgarian NMR Centre, Institute of Organic Chemistry with Centre of Phytochemistry, Bulgarian Academy of Sciences, 1113 Sofia, Bulgaria; 2Laboratory Organic Synthesis and Stereochemistry, Institute of Organic Chemistry with Centre of Phytochemistry, Bulgarian Academy of Sciences, 1113 Sofia, Bulgaria

**Keywords:** floral, honeydew and adulterated honey, optical rotation, ^13^C NMR, botanical origin

## Abstract

The controversial question of whether optical rotation data can be used to distinguish floral from honeydew honey was investigated. Specific optical rotation angles were determined for 41 honey samples, including floral, honeydew, and adulterated honey, indicating that moderate to high positive optical rotation angles were found for all adulterated samples measured. A strong correlation between the sugar profile and the specific optical rotation angle of honey was confirmed, and a method based on ^13^C NMR metabolomics was proposed to calculate specific optical rotation angles with good correlation with the experimental values. The results indicate that optical rotation is not a reliable method for distinguishing the origin of honey but could indicate adulteration.

## 1. Introduction

Honey is the oldest natural sweetener known to mankind since ancient times, with bee products being used at least 8000 years ago [1]. In every culture, there is evidence of the use of honey as a food source and as a substance used in religious, magical and therapeutic ceremonies. “Honey is the natural sweet substance produced by *Apis mellifera* bees from the nectar of plants or from secretions of living parts of plants or excretions of plant-sucking insects on the living parts of plants, which the bees collect, transform by combining with specific substances of their own, deposit, dehydrate, store and leave in honeycombs to ripen and mature” [2]. Honey continues to be widely used in traditional and veterinary medicine, pharmacy and cosmetics. It is known for its antioxidant, antibacterial, anticancer and anti-inflammatory effects. Natural honey prevents diabetes, heart and eye diseases [3]. It is used against burns, diarrhea, gastroenteritis, hepatitis, tuberculosis, asthma, and helps in a quick recovery from COVID-19 [4,5,6].

The number of food frauds has increased manifold over the past two centuries, posing a serious problem for the economy, industry, and human health. The counterfeiting of various foods worldwide every year leads to economic losses of USD 10–15 billion, affecting more than 10% of all food sold [7]. The number of adulterated goods continues to increase, despite the annual fight against food fraud [8]. Certain studies show that the financial advantages of this type of activity are similar to those of cocaine trafficking [9]. The exact number is unknown, as most counterfeit foods do not have a direct impact on health and consumers often do not notice product quality issues.

Honey has been one of the most commonly counterfeited foods in the world over the last decade [10]. A number of classical and advanced analytical methods have been developed and used for authentication and detection of adulteration. The most popular are presented in Figure 1.

Pollen analysis and the combination of the physicochemical characteristics of honey—moisture and sugar content (main reducing saccharides and sucrose), electrical conductivity, 5-hydroxymethylfurfural amount, enzyme activity and total acidity are often used to determine origin and/or quality in some countries [11,12]. Most of these methods are time-consuming, require trained personnel and special reagents and/or equipment, and are not reliable enough to detect honey diluted with some types of syrups [13]. There is a growing need for reliable identification and quantification of the main organic substances in different honey types, especially for different honeydew types. A free and reliable database of honeys from various botanical, entomological, and geographical origins has still not been published; information on the components that differentiate them is sparse. Despite recent extensive research on its composition, there are still no officially accepted methods for the analysis and proof of its quality and authenticity.

Honey has the property of optical rotation, i.e., its composition includes optically active substances that rotate the polarized light at a certain angle [α]. The specific angle of honey depends on the quantity and ratio between the main sugar constituents—the levorotatory fructose (−) and the dextrorotatory glucose (+). However, different low quantity organic components in honey with large positive or negative rotation angles could significantly contribute to its specific rotation. It has been observed that a number of honeydew honeys are dextrorotatory, differing from nectar honeys, which have negative specific angles [14]. One possible explanation is the lower fructose content in some honeydew honeys, as well as the presence of di- and trisaccharides with large positive angles. Measurement of the specific angle has been reported for distinction of honeydew from nectar honey in studies from Italy [15], Poland [16], Bulgaria [17], Greece and the UK [18]. However, the method is not recognized by the IHC due to lack of sufficiently reliable data [19]. Primorac et al. found several honeydew honeys from North Macedonia and Croatia with negative angles of rotation, recommending further research in this area [20]. Recently, Serrano et al. studied key factors affecting specific optical rotation determination in honey and suggested introducing an uncertainty interval of −5 to +5 due to the measured overlap between the optical rotation values of nectar and honeydew honey [21].

We recently used NMR metabolomics for discrimination of botanical, geographical and entomological origin of honey [22,23,24]. This methodology allows the detailed determination of the sugar profile of honey, as well as an assessment of the amounts of a number of other organic components such as organic and amino acids, alcohols, 5-hydroxymethylfurfural (HMF) and a small quantity of unidentified compounds. In the present study, we applied the idea to match the sugar profile and the specific optical rotation, performed initially using HPLC [25]. NMR is superior for this task since it allows one to quantitate the 21 main sugars in honey, providing better correlations thus allowing more insight to be gained into the possibilities of using specific optical rotation for analytical purposes to provide the distinction of different honey types.

## 2. Results

### 2.1. Measured Specific Optical Angles

The established methodology for the determination of specific optical rotation of honey, involving complex sample preparation—dissolving in water, adding Carrez I and Carrez II solutions, filtering and standing for 24 h was applied for 41 honey samples and one sample of pine cone jam (**jm**). The measured minimum, maximum and average values of the specific angles of rotation ${\left[\alpha \right]}_{D}^{20}$ for the individual types of honey, as well as the number (*n*) of samples for the different types measured, are presented in Table 1. For pine jam an angle of +22.7 was determined.

Comparison with literature data showed similarities in angle size for nectar honey species, regardless of exact plant and geographic origin, but large differences were observed for many honeydew samples. Special attention should be paid to the large spread in the values of adulterated samples, containing different quantities of sugar syrups.

### 2.2. Calculation of Specific Optical Rotation of Honey/Jam Based on Its Sugar Profile

Based on the proposed dependency of the specific optical rotation on its sugar profile [20], we developed a method for calculation of the specific angle of rotation of honey or jam ${\left[\alpha \right]}_{\mathrm{hon}}$. The method uses the concentrations of 21 identified saccharides determined from their ^13^C NMR spectra of the individual honey samples and the ${\left[\alpha \right]}_{D}^{20}$ for the individual sugars, described in the literature. The proposed formula to perform the calculation is presented below, where ${\%}_{\mathrm{sug}}$ and ${\left[\alpha \right]}_{\mathrm{sug}}$ are the percentage and the specific optical angle of each saccharide in the honey:(1)$${\left[\alpha \right]}_{\mathrm{hon}}=\sum_{n=21}\left(\frac{{\%}_{\mathrm{sug}\left(1\right)}\times {\left[\alpha \right]}_{\mathrm{sug}\left(1\right)}}{100}\right)+\ldots +\left(\frac{{\%}_{\mathrm{sug}\left(n\right)}\times {\left[\alpha \right]}_{\mathrm{sug}\left(n\right)}}{100}\right)$$

In Table 2, taking the example of chestnut honey **ch8,** the calculation of the specific optical angle is presented using the known literature data for the individual saccharides.

## 3. Discussion

Table 2 indicates the positive angle values primarily measured for dilute honey. Apart from these samples (**dl1**–**dl7**), only two other honeydew honeys—**cf13** from Italy, **ch11** from Bulgaria, and one nectar honey **ts25** from Bulgaria—have a positive rotation angle. The other data differ from those described for Italian honey—from +6.0 to +29.7 [15], for Polish honey (+0.7 to +9.6) [14,16] and for Bulgarian honey (+2.9 to +5.5) [17]. Several probable reasons for the differences in the angles measured in these studies and our results could be listed—different types of sucking insects that produce honeydew, different plant varieties, inaccuracies in the research or analysis of diluted honeydew. However, our measured values for oak, conifer and chestnut honeydew were close to the angles for honeydew from Croatia (−7.7 to +18.2), North Macedonia (−7.8 to +7.5) [20] and for Turkish oak honey from the region of Thrace (−0.6 to +1.6) [26].

The negative specific angle of linden honey from Italy [15], rapeseed honey from Poland [14], as well as acacia honey from Italy, Bulgaria and Croatia had similar values to those found in Table 1. Despite the great heterogeneity in the composition of polyfloral honey, the angles measured were similar to those described in Polish (from −17.1 to −15.0) [16] and Bulgarian studies (from −19.7 to −9.9) [17]. A difference was observed in thyme honey (sample **tm27**), with a higher value than that of Italian honey of identical plant origin [15], as well as in Latvian linden honey [27]. Visualization of the measured data spread for the individual measured types of honey and jam is presented graphically in Figure 2.

It is clear from these data that measurement of the specific rotation angle does not provide reliable information to distinguish honey by botanical origin. A clear distinction of diluted honey having ${\left[\alpha \right]}_{D}^{20}$ values from +7.2 to +89.2 from nectar/mixed honey having a not overlapped interval from −35.2 to +2.5. Based on these results, the method will be useful for detecting adulterated honey. In order to check the influence of the different sugar components on the size of the rotation angle, we performed a correlation analysis that compared the quantitative data for the identified saccharides determined from the ^13^C NMR spectra with the specific rotation angle of the 42 samples analyzed. The data were standardized by z-transformation. The Pearson correlation coefficient (r) showing the relationship between the optical angle and individual saccharides is presented in Table 3. A detailed correlation matrix with the Pearson coefficient (r) and the effects between all studied sugars is presented in Appendix A.

The data in Table 3 confirm the results of Primorac’s research [20] on the different influences of the individual sugars on the honey specific angle. The main negative angle influence was from fructose, while maltose and maltotriose showed high positive contributions to the angle. These saccharides are usually present in small quantities in honey, but are quite abundant in sugar syrups, commonly used for honey adulteration. Some other common sugars in honey such as turanose and αβ-trehalose also have a non-negligible influence. The proposed method for the calculation of the specific rotation angle of honey or jam, presented in Equation (1), allows in depth analysis of the possible analytical applications of the specific rotation angle.

The angles of the analyzed 41 samples of honey and 1 sample of jam were calculated using Equation 1. The calculated [α]cal values were compared with the experimentally determined [α]exp. Both values were quite close, as shown in Figure 3, and indicated a good linear dependence with a correlation coefficient higher 0.9, as presented in Figure 4.

Despite of the high level of similarity observed between the theoretically calculated and experimentally determined angles, some differences were also observed. The NMR analysis showed that different individual types of honey—linden, rapeseed, thyme and polyfloral—contained different amounts of unidentified components U1-U16, that are statistically important for discrimination. The observed difference between [α]exp and [α]cal varied from 3 to 15%. The best results were obtained for the pine cone jam, that had a simple and fully established sugar profile, the difference being ${\left[\alpha \right]}_{D}^{20}$ of 1.5 (7%). If the concentration of the unidentified substances is above 1.37%, the difference can be more than 20%. To analyze these differences, the figures of the linear dependences between [α]exp and [α]cal were drawn for the individual classes of honey with more than two samples and are presented in Figure 5.

The weakest linear dependence was observed for oak honey. The samples forming this class of honey could be divided into two groups according to the size of the calculated angle. For samples **ok15**–**ok21**, the calculated angle was larger than the experimentally determined one, and for samples **ok22**–**ok24**, it was smaller. The samples with [α]cal > [α]exp originated from Bulgaria and the other three from North Macedonia and Romania. The coefficient R^2^ for the linear correlation for oak honey increased from 0.2802 to 0.6864 after exclusion of these three samples. This indicates that not only the botanical but also the geographical origin of the samples may have an influence on the specific rotation angle, possible reasons being the presence of different identified or unidentified chiral substances due to different plant varieties in the corresponding geographical region. For the honeydew honeys, this is even more important since also the nature of the plant-sucking insects may differ leading to the involvement of different di- or trisaccharides in the mature honeys.

The developed method provides good accuracy for calculating the specific angle for honey and jam with fully established sugar profile. The differences between the calculated and experimental angles for some honey types may be explained by the presence of other undetermined chiral compounds with high specific rotation. Further analyses, especially for the relatively poorly researched composition of honeydew honeys, are needed to achieve a higher accuracy of the method to detect the statistically significant components, especially in honeydew oak and chestnut honey.

## 4. Materials and Methods

### 4.1. Honey Samples

Thirty-four randomly selected floral and honeydew honey samples with different botanical and geographical origins were analyzed in this study. Additionally, seven adulterated honeys and one pine cone jam were investigated analogously. The exact origins of the measured samples are presented in Appendix A.

### 4.2. Sample Preparation

For determination of the specific optical rotation angle of honey, Carrez I (1.06 g of K_4_Fe(CN)_6_·3H_2_O, dissolved in 10 mL distilled water) and Carrez II (2.4 g of Zn(CH_3_COO)_2_·2H_2_O and 0.3 g of anhydrous acetic acid, dissolved in 10 mL distilled water) solutions were prepared. A measure of 1.2 g of the honey sample was dissolved in distilled water and poured into a 10 mL volumetric flask. Then, 1 mL of Carrez I and 1 mL of Carrez II were added, and after the addition of each of the solutions, the mixture was stirred for 30 s. Resting for 24 h resulted in a precipitate that was filtered and the solution transferred to the polarimeter quartz tube. The optical rotation values of several samples selected on a random principal were re-measured within a week. The same values were obtained showing that no sugar decomposition had taken place when using the particular sample preparation.

### 4.3. Determination of the Specific Optical Rotation (SOR)

Measurement of angular rotation (α**_D_**) was performed at 20 °C on a Jasco P-2000 polarimeter (Tokyo, Japan) equipped with a deuterium lamp. Specific optical rotation of the studied honeys and jam were determined according to the Harmonized Methods of the European Honey Commission using the formula α**_D_** = α × 100/l × p, where α = angular rotation found, l = length in decimeters of the polarimeter tube (0.5 dm used), p = grams of dry matter taken. The ${\left[\alpha \right]}_{D}^{20}$ are given in deg·cm^3^·g^−1^·dm^−1^, concentration in g·cm^−3^ [28].

### 4.4. NMR Spectroscopy

NMR spectra were acquired on a Bruker Avance II+ 600 and Bruker NEO 600 spectrometers (Biospin GmbH, Rheinstetten, Germany) at 300.0 ± 0.1 K using BBO/Prodigy probeheads. Standard parameters for broad band decoupled ^13^C NMR spectra were used—pulse sequence zgdc30, pulse width 30°, spectral width 238 ppm, 64 K data points, 8/4 K scans, acquisition time 0.90 s, and relaxation delay 1.05 s. The signal of α-fructofuranose at 104.34 ppm was used as internal reference corresponding to −2.82 ppm for the ^13^C TSP signal. Signal assignments and semiquantitative analysis were achieved as described previously using 1D and 2D spectra and literature data. Quantitation of the sugar components was made from ^13^C-NMR signals of two monosaccharides (glucose, fructose), 13 disaccharides (sucrose, kojibiose, α,α- and α,β-trehalose, trehalulose, maltose, isomaltose, maltulose, isomaltulose, nigerose, leucrose, turanose, gentiobiose), 6 trisaccharides (raffinose, melezitose, 1-kestose, panose, erlose, maltotriose). The detailed NMR analysis allowed us to quantify, in addition to sugars, several identified and 16 unidentified organic components that are important for the characterization of honey [23,29].

### 4.5. Correlation Analysis

Correlation analysis to determine the effect of the individual saccharides on the optical angle of honey was carried out using Pearson correlation coefficient (r) in Excel. The formula for determining the Pearson coefficient is:$$r_{xy}=\frac{n\sum x_{i}y_{i}-\sum x_{i}\sum y_{i}}{\sqrt{n\sum x_{i}^{2}-{\left(\sum x_{i}\right)}^{2}}\sqrt{n\sum y_{i}^{2}-{\left(\sum y_{i}\right)}^{2}}}$$
where *r_xy_* is the Pearson correlation coefficient between variables of specific optical rotation of the individual sugars (*x*) and corresponding data for honey (*y*), n is the number of objects (42 samples), and *x_i_* and *y_i_* represent values of *x* and *y* for observation *i*.

## 5. Conclusions

The work presented shows that care should be taken when moderate to high positive values are measured for the specific optical rotation angles of honey due to the high probability of encountering adulteration. Optical rotation data cannot be used to distinguish unambiguously floral from honeydew honey. The sugar profile and chemical composition of chiral compounds is highly dependent on botanical/geographical origin. NMR provides a useful method to control the specific optical rotation angle against the chemical profile, with a major part—the sugar profile.

## Figures and Tables

**Figure 1 molecules-27-08916-f001:**
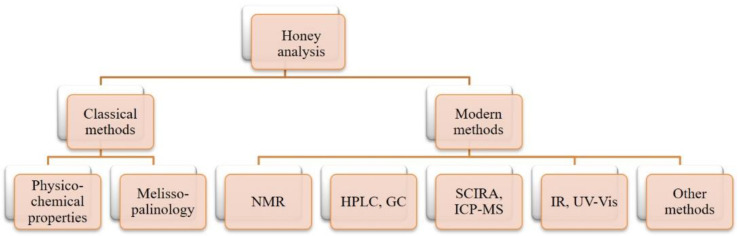
Different methods for honey analysis and authentication.

**Figure 2 molecules-27-08916-f002:**
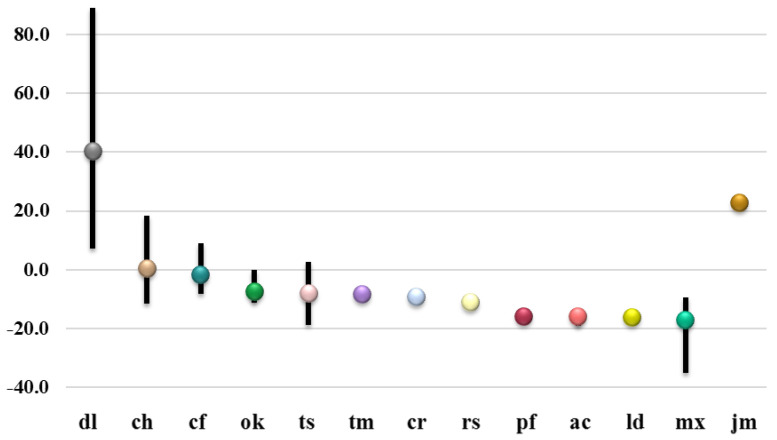
Magnitude of specific rotation angle for different honey types—diluted (dl), chestnut (ch), conifer (cf), oak (ok), thistle (ts), thyme (tm), coriander (cr), rapeseed (rs), polyfloral (pf), acacia (ac), linden (ld), mixed (mx), and pine cone jam (jm).

**Figure 3 molecules-27-08916-f003:**
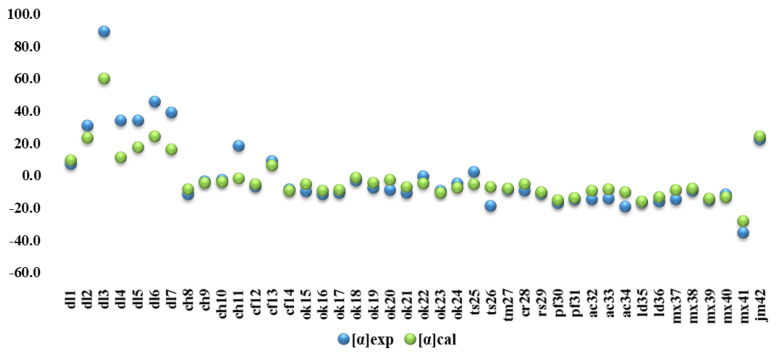
Calculated [α]cal and experimental [α]exp values of specific angle of 41 samples of honey and one sample of jam.

**Figure 4 molecules-27-08916-f004:**
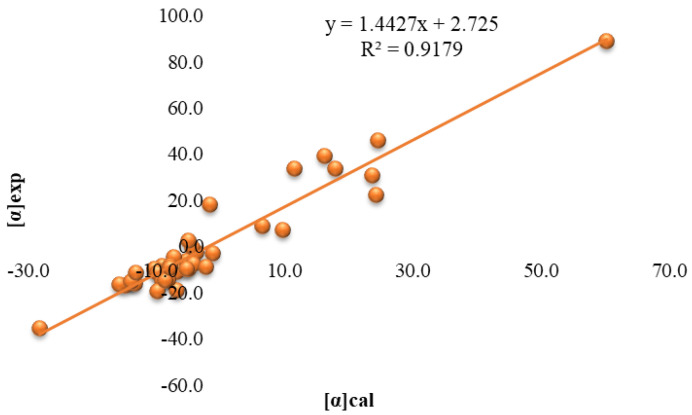
Linear dependence between [α]cal and [α]exp.

**Figure 5 molecules-27-08916-f005:**
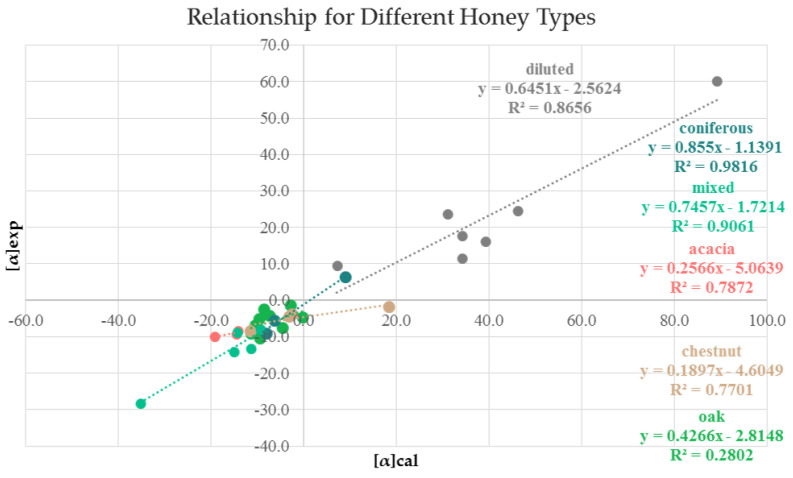
Linear relationship between experimentally determined [α]exp and calculated [α]cal optical rotation angle for different honey types.

**Table 1 molecules-27-08916-t001:** Minimum (min), maximum (max) and average (avg) specific angle values of honey of different botanical origin, including adulterated (diluted) samples.

${\left[\alpha \right]}_{D}^{20}$	Diluted (*n* = 7) *	Chestnut (*n* = 4)	Coniferous (*n* = 3)	Oak (*n* = 10)	Thistle (*n* = 2)	Thyme (*n* = 1)
*min* *max*	+7.2 +89.2	−11.5 +18.4	−8.1 +9.0	−11.3 −0.2	−18.8 +2.5	
*avg*	*+40.2*	*+0.3*	*−1.8*	*−7.5*	*−8.3*	*−8.3*
${\left[\alpha \right]}_{D}^{20}$	**Coriander** **(*n* = 1)**	**Rapeseed** **(*n* = 1)**	**Polyfloral** **(*n* = 2)**	**Acacia** **(*n* = 3)**	**Linden** **(*n* = 2)**	**Mixed** **(*n* = 5)**
*min* *max*			−16.9 −14.8	−19.2 −14.1	−16.3 −15.8	−35.2 −9.3
*avg*	*−9.4*	*−11.2*	*−15.9*	*−16.0*	*−16.1*	*−17.0*

* *n* is the number of samples measured for the individual honey types.

**Table 2 molecules-27-08916-t002:** Example for calculation of specific optical rotation of honey using the sugar profile.

Saccharides in Honey	Individual Angle [α]_sug_	Sugar Quantity (%) g/100 g	Contribution to the OR
Glucose (G)	52.70	28.92	*15.20*
Fructose (F)	−92.40	37.48	*−34.60*
Gentiobiose (Gb)	10.00	0.00	*0.00*
Sugar (Su)	66.50	0.25	*0.20*
Isomaltose (IMa)	120.00	1.65	*2.00*
Isomaltulose (IMu)	97.20	0.00	*0.00*
Kojibiose (Kb)	133.00	0.54	*0.70*
Leucrose (Lu)	−6.80	0.15	*0.00*
Maltose (Ma)	130.40	1.87	*2.40*
Maltulose (Mu)	64.00	1.79	*1.10*
Nigerose (Ng)	120.00	0.59	*0.70*
ααTrehalose (Tr)	199.00	0.00	*0.00*
αβTrehalose	82.00	0.31	*0.30*
Trehalulose (Tru)	50.00	0.61	*0.30*
Turanose (Nu)	75.00	2.09	*1.60*
Erlose (Er)	121.80	0.71	*0.90*
1−Kestose (1−Ks)	28.00	0.18	*0.10*
Maltotriose (Mr)	160.00	0.00	*0.00*
Melezitose (Mz)	88.20	0.00	*0.00*
Panose (Pa)	154.00	0.31	*0.50*
Raffinose (Rf)	101.00	0.39	*0.40*
	Specific optical rotation angle of honey:		*−8.3*

**Table 3 molecules-27-08916-t003:** Pearson’s coefficient (r) determining the effect of individual saccharides on the optical angle of honey.

${\left[\alpha \right]}_{D}^{20}$	**G**	**F**	**Gb**	**IMa**	**IMu**	**Kb**
	*0.09* *	**−0.89** **	*−0.22*	*−0.20*	*−0.19*	*−0.54*
${\left[\alpha \right]}_{D}^{20}$	**Ma**	**Mu**	**Ng**	**Su**	**ααTr**	**αβTr**
	**0.88**	*−0.44*	*−0.50*	*0.15*	*−0.16*	−0.58
${\left[\alpha \right]}_{D}^{20}$	**Tu**	**Er**	**1-Ks**	**Mr**	**Mz**	**Pa**
	−0.62	*−0.35*	*−0.21*	**0.89**	*−0.07*	*−0.08*

* Data without statistical significance at the *p* > 0.05 level are marked in italics. ** Very high positive (r > 0.8) and negative correlations (r < −0.8) are shown in bold.

## Data Availability

The data of the current study is available from the corresponding authors on reasonable request.

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
