# Peer review of "Optical Rotation—A Reliable Parameter for Authentication of Honey?"

_molecules, 2022, doi:10.3390/molecules27248916_

Round 1

Reviewer 1 Report

The authors investigated whether optical rotation data can be used to distinguish floral from honeydew honey.  They collected 34 randomly selected floral and honeydew honey samples with different botanical and geographical origins and they found that optical rotation data cannot be used to distinguish floral from honeydew honey unless the honey types studied have been fully studied.  the study is of moderate importance as there are several methods that have been used to distinguish the botanical and geographical origin of honey. I have some comments as follows:

1- In the abstract:  the authors need to provide a conclusion of their findings, for example, optical rotation data cannot be used to distinguish floral from honeydew honey unless the honey types studied have been fully studied.

2- Line 21-22: Please provide relevant references.

3- Line 31-32: Please cite relevant and recent articles here:

Al Naggar, Y., Giesy, J. P., Abdel-Daim, M. M., Ansari, M. J., Al-Kahtani, S. N., & Yahya, G. (2021). Fighting against the second wave of COVID-19: Can honeybee products help protect against the pandemic?. Saudi journal of biological sciences, 28(3), 1519-1527.

Shaldam, M. A., Yahya, G., Mohamed, N. H., Abdel-Daim, M. M., & Al Naggar, Y. (2021). In silico screening of potent bioactive compounds from honeybee products against COVID-19 target enzymes. Environmental Science and Pollution Research, 28(30), 40507-40514.

4- Line 41-43: This sentence needs to be supported by references. Also, many studies already tried chemical and pollen analysis methods and these should be mentioned then the author highlights the disadvantages of these methods and then argue for more reliable methods (optical rotation).

https://pubs.rsc.org/en/content/articlehtml/2021/ra/d1ra00069a

https://www.sciencedirect.com/science/article/pii/S0308814619317194

https://www.tandfonline.com/doi/abs/10.1080/00218839.2020.1720950

5- Line 51: low instead of law

6- Line 58: Provide references here as well for Greece and Great Britain.

7- line 94: [] ??

Author Response

Response to reviewer 1 comments

We greatly appreciate the reviewers' insightful and helpful comments on our manuscript. It has been revised according to the recommendations of the reviewers. We have added a short paragraph listing alternative analytical methods. Some clarifications and additional literature sources have been added. All spelling and grammar errors pointed out by reviewers have been corrected. As well as other found.

 Below the point by point answers to reviewer 1 recommendations are presented.

1- In the abstract: the authors need to provide a conclusion of their findings, for example, optical rotation data cannot be used to distinguish floral from honeydew honey unless the honey types studied have been fully studied.

      A conclusion statement has been added to the abstract.

2- Line 21-22: Please provide relevant references.

      The reviewer's suggestion is quite in place and we added the corresponding reference.

3- Line 31-32: Please cite relevant and recent articles here:

Al Naggar, Y., Giesy, J. P., Abdel-Daim, M. M., Ansari, M. J., Al-Kahtani, S. N., & Yahya, G. (2021). Fighting against the second wave of COVID-19: Can honeybee products help protect against the pandemic?. Saudi journal of biological sciences, 28(3), 1519-1527.

Shaldam, M. A., Yahya, G., Mohamed, N. H., Abdel-Daim, M. M., &Al Naggar, Y. (2021). In silico screening of potent bioactive compounds from honeybee products against COVID-19 target enzymes. Environmental Science and Pollution Research, 28(30),40507-40514.

      We have added two review articles from this year citing the above references.

4- Line 41-43: This sentence needs to be supported by references. Also, many studies already tried chemical and pollen analysis methods and these should be mentioned then the author highlights the disadvantages of these methods and then argue for more reliable methods (optical rotation).

https://pubs.rsc.org/en/content/articlehtml/2021/ra/d1ra00069a

https://www.sciencedirect.com/science/article/pii/S0308814619317194

https://www.tandfonline.com/doi/abs/10.1080/00218839.2020.1720950

       In line with the reviewer’s recommendations, we added a new short paragraph that lists the most used analytical methods for honey authentication, accompanied with a suitable figure. Since the main subject of the manuscript is the predictive power of optical rotation we do not discuss in any detail the broad subject of other methods for honey authenticity determination. 

5- Line 51: low instead of law

       Corrected

6- Line 58: Provide references here as well for Greece and Great Britain.

      To correct the absence of literature sources for these countries we cite the fundamental work “Honey quality and international regulatory standards: review by the International Honey Commission” that states “The measurement of specific rotation is currently used in Greece, Italy and the UK to distinguish between blossom and honeydew honeys”.

7- line 94: [] ??

       Corrected

Reviewer 2 Report

The article presents a study of the possibility of using optical rotation to determine the type of honey.  Despite the fact that the final parameters for the application of the method for assessing the type of honey have not been identified, important scientific data and relationships have been obtained that expand the currently available ideas in this field of knowledge. A strong 14 correlation between the sugar profile and the specific optical rotation angle of honey was confirmed. The article will be useful to researchers and producers working with honey.  Potentially, extending the findings could lead to the development of a practical method for typing and adulteration of honey.

Despite the good quality of the manuscript, there are a number of suggestions for its improvement.

1) Lines 30-31. The statement "Natural honey prevents diabetes, heart and eye diseases."  requires relevant references.

2) Lines 42-43. Despite the fact that the statement "there are still no officially accepted methods for analysis and prove of its quality and authenticity" is generally true, it is worth noting the possibility of assessing the quality and falsification of honey in certain geographical areas based on a combination of parameters such as diastase number,  sucrose content, reaction to hydroxymethylfurfural.

3) There are no red and blue colors in the Table 3, as it is stated in the Lines 141-142.

4) Lines 246-247. The sentence "Optical rotation data cannot be used to distinguish floral from honeydew honey unless the honey types have been fully studied."  should be rephrased.

5) In the conclusion or discussion, it is worth suggesting the possibility of using optical rotation to assess the quality and fakes of honey in specific geographic regions, where there will not be such a strong spread between the influencing parameters.

Author Response

Response to reviewer 2 comments

We greatly appreciate the reviewers' insightful and helpful comments on our manuscript. It has been revised according to the recommendations of the reviewers. We have added a short paragraph listing alternative analytical methods. Some clarifications and additional literature sources have been added. All spelling and grammar errors pointed out by reviewers have been corrected. As well as some other found.

Below the point by point answers to reviewer 2 recommendations are presented:

1) Lines 30-31. The statement "Natural honey prevents diabetes, heart and eye diseases." requires relevant references.

       The relevant reference is added in the wright place.

2) Lines 42-43. Despite the fact that the statement "there are still no officially accepted methods for analysis and prove of its quality and authenticity" is generally true, it is worth noting the possibility of assessing the quality and falsification of honey in certain geographical areas based on a combination of parameters such as diastase number, sucrose content, reaction to hydroxymethylfurfural.

       In line with the reviewer’s recommendations, we added a new short paragraph that lists the most used analytical methods for honey authentication, accompanied with a suitable figure. Since the main subject of the manuscript is focused on the predictive power of optical rotation we do not discuss in any detail the broad subject of other methods for honey authenticity determination. 

3) There are no red and blue colors in the Table 3, as it is stated in the Lines 141-142.

       When submitting the manuscript, we did not realize that the tables do not use colours, in the revised version we provide indication of the most important values in bold.

4) Lines 246-247. The sentence "Optical rotation data cannot be used to distinguish floral from honeydew honey unless the honey types have been fully studied." should be rephrased.

       The sentence is shortened. Since our data do not unambiguously indicate the possible use of optical rotation for distinction of floral/honeydew honey, we decided not to speculate such possibilities, although in special cases they could be possible.  

5) In the conclusion or discussion, it is worth suggesting the possibility of using optical rotation to assess the quality and fakes of honey in specific geographic regions, where there will not be such a strong spread between the influencing parameters.

        See answer to question 4)

Round 2

Reviewer 1 Report

All comments were successfully addressed by the authors.